# Modulation of sleep-courtship balance by nutritional status in *Drosophila*

**José M Duhart[1], Victoria Baccini[1], Yanan Zhang[1], Daniel R Machado[1,2,3], Kyunghee Koh[1]***

[1]Department of Neuroscience, Farber Institute for Neurosciences, Thomas Jefferson University, Philadelphia, United States; [2]Life and Health Sciences Research Institute (ICVS), School of Health Sciences, University of Minho, Braga, Portugal; [3]ICVS/3B's, PT Government Associate Laboratory, Braga, Portugal

**Abstract** Sleep is essential but incompatible with other behaviors, and thus sleep drive competes with other motivations. We previously showed *Drosophila* males balance sleep and courtship via octopaminergic neurons that act upstream of courtship-regulating P1 neurons (Machado et al., 2017). Here, we show nutrition modulates the sleep-courtship balance and identify sleep-regulatory neurons downstream of P1 neurons. Yeast-deprived males exhibited attenuated female-induced nighttime sleep loss yet normal daytime courtship, which suggests male flies consider nutritional status in deciding whether the potential benefit of pursuing female partners outweighs the cost of losing sleep. Trans-synaptic tracing and calcium imaging identified dopaminergic neurons projecting to the protocerebral bridge (DA-PB) as postsynaptic partners of P1 neurons. Activation of DA-PB neurons led to reduced sleep in normally fed but not yeast-deprived males. Additional PB-projecting neurons regulated male sleep, suggesting several groups of PB-projecting neurons act downstream of P1 neurons to mediate nutritional modulation of the sleep-courtship balance.

*For correspondence:
kyunghee.koh@jefferson.edu

**Competing interests:** The authors declare that no competing interests exist.

## Introduction

Sleep is observed in every animal species studied in detail (*Anafi et al., 2019*), underscoring its importance for fitness. A widely accepted framework for understanding sleep regulation, called the two-process model, proposes that sleep is controlled by the circadian and homeostatic processes that convey information about the time of day and sleep drive, respectively (*Borbély, 1982*). However, since sleep prevents the execution of other critical behaviors such as feeding and mating, sleep is also influenced by motivational factors such as hunger and sex drive. For instance, sleep is suppressed by starvation in both rats and fruit flies, likely to allow the animal to forage for food (*Jacobs and McGinty, 1971*; *Keene et al., 2010*). Similarly, female sleep is reduced upon mating, presumably for egg-laying purposes (*Garbe et al., 2016*; *Isaac et al., 2010*). Recently, we and others have shown that sleep is suppressed in favor of courtship when male flies are paired with females (*Beckwith et al., 2017*; *Machado et al., 2017*), demonstrating a competition between sleep and sex drive.

In addition to sleep and sex drive, both general and nutrient-specific hunger are important modulators of behavior. For instance, yeast deprivation in *Drosophila* alters food choice in favor of high-protein food over the normal preference for high-carbohydrate food (*Ribeiro and Dickson, 2010*). In addition, yeast provides essential nutrients for proper larval development (*Anagnostou et al., 2010*; *Robertson, 1960*), and the amount of yeast in the female diet correlates with the number of eggs laid (*Lin et al., 2018*). Although the effects of dietary yeast on male reproduction are relatively modest (*Zajitschek et al., 2013*; *Fricke et al., 2008*), we hypothesized that it may have a stronger influence on the choice between sleep and reproductive behavior in male flies.

Whereas a number of neuronal populations that regulate sleep or courtship have been identified (*Artiushin and Sehgal, 2017*; *Ellendersen and von Philipsborn, 2017*), only a few neuronal populations regulating both behaviors (i.e. sleep and courtship) are known. Among these, P1 neurons, which express the Fruitless[M] (Fru[M]) transcription factor and play a critical role in courtship behavior (*Clyne and Miesenböck, 2008*; *Kimura et al., 2008*; *Manoli et al., 2005*; *Stockinger et al., 2005*), are also involved in male sleep regulation (*Beckwith et al., 2017*; *Chen et al., 2017*; *Machado et al., 2017*). P1 neurons are known to receive male-specific arousal signal from octopaminergic MS1 neurons (*Machado et al., 2017*) and act both upstream and downstream of DN1 clock neurons (*Chen et al., 2017*) to regulate the sleep-courtship balance. However, how P1 neurons communicate with downstream sleep circuits remains unknown.

Here, we demonstrate that the sleep-courtship balance in male flies is affected by yeast deprivation in *Drosophila* and identify the protocerebral bridge (PB) as an arousal center acting downstream of P1 neurons. Yeast-deprived male flies exhibited attenuated female-induced nighttime sleep loss relative to normally fed males. In contrast, yeast deprivation did not impair the ability of males to court during the day, suggesting that dietary yeast affects the sleep-courtship balance rather than courtship per se. Using the trans-Tango trans-synaptic tracing technique (*Talay et al., 2017*), we identified a pair of dopaminergic neurons projecting to the protocerebral bridge (DA-PB) as neurons acting downstream of the P1 cluster. Calcium imaging confirmed a functional connection between the two groups of neurons. Furthermore, activation of DA-PB neurons led to sleep suppression in normally fed but not yeast-deprived males. Through a screen of PB-arborizing neurons, we identified additional neurons that regulate sleep specifically in males. We conclude that male sleep suppression by female cues is strongly affected by nutritional conditions and that P1, DA-PB, and additional PB-projecting neurons form a neural circuit for integrating sleep and sex drives in males.

## Results

### Yeast deprivation modulates the balance between sleep and courtship

We first examined whether nutritional status affects the balance between sleep and courtship in *Drosophila* males. Based on previous findings that 7 days of sucrose-only diet alters the internal state of male flies (*Ribeiro and Dickson, 2010*), we assessed sleep in male-male (MM) or male-female (MF) pairs after 7 days of sucrose-only diet. We fed groups of control (iso31) flies 5% sucrose food or normal food (standard food for *Drosophila* maintenance, see Materials and methods) for 6 days, loaded the flies into tubes containing 5% sucrose in Male-Male (MM) or Male-Female (MF) pairs, and assessed sleep the next day using the single-beam *Drosophila* Activity Monitor (DAM) system (*Figure 1A*). As previously reported (*Beckwith et al., 2017*; *Machado et al., 2017*), under the normal food condition, MF pairs showed a marked reduction in sleep compared with MM pairs (*Figure 1B–D*). Strikingly, yeast-deprived MF pairs showed increased nighttime sleep (i.e. reduced sleep suppression) compared with normally fed MF pairs (*Figure 1B,D*). In contrast, yeast deprivation resulted in a small decrease in sleep in MM pairs and individual males, and had little effect on sleep in individual females (*Figure 1C,D*, *Figure 1—figure supplement 1*). These results demonstrate that the effects of yeast deprivation on sleep depend on the social context.

To rule out any contribution of the female nutritional status to the MF sleep, we paired yeast-deprived males with females that were kept in normal food until they were placed in tubes containing 5% sucrose food for sleep assay. Sleep in MF pairs with yeast-deprived males was independent of the nutritional status of the females in the pair (*Figure 1—figure supplement 2*), indicating that the effects of nutrition on MF sleep are due to its effects on male behavior. Since the present study is focused on the effects of male nutrition on male behavior, normally fed females were used in MF pairs in subsequent experiments.

To control for the possibility that the single-beam DAM system with a single infrared detector missed small movements in yeast-deprived flies, we examined the effects of yeast deprivation using the multi-beam DAM system containing 17 infrared detectors. The multi-beam DAM system has two modes of analyzing movements: 'moves,' which include only inter-beam movements (i.e. movements between beams), and 'counts,' which also include intra-beam movements (i.e. local movements

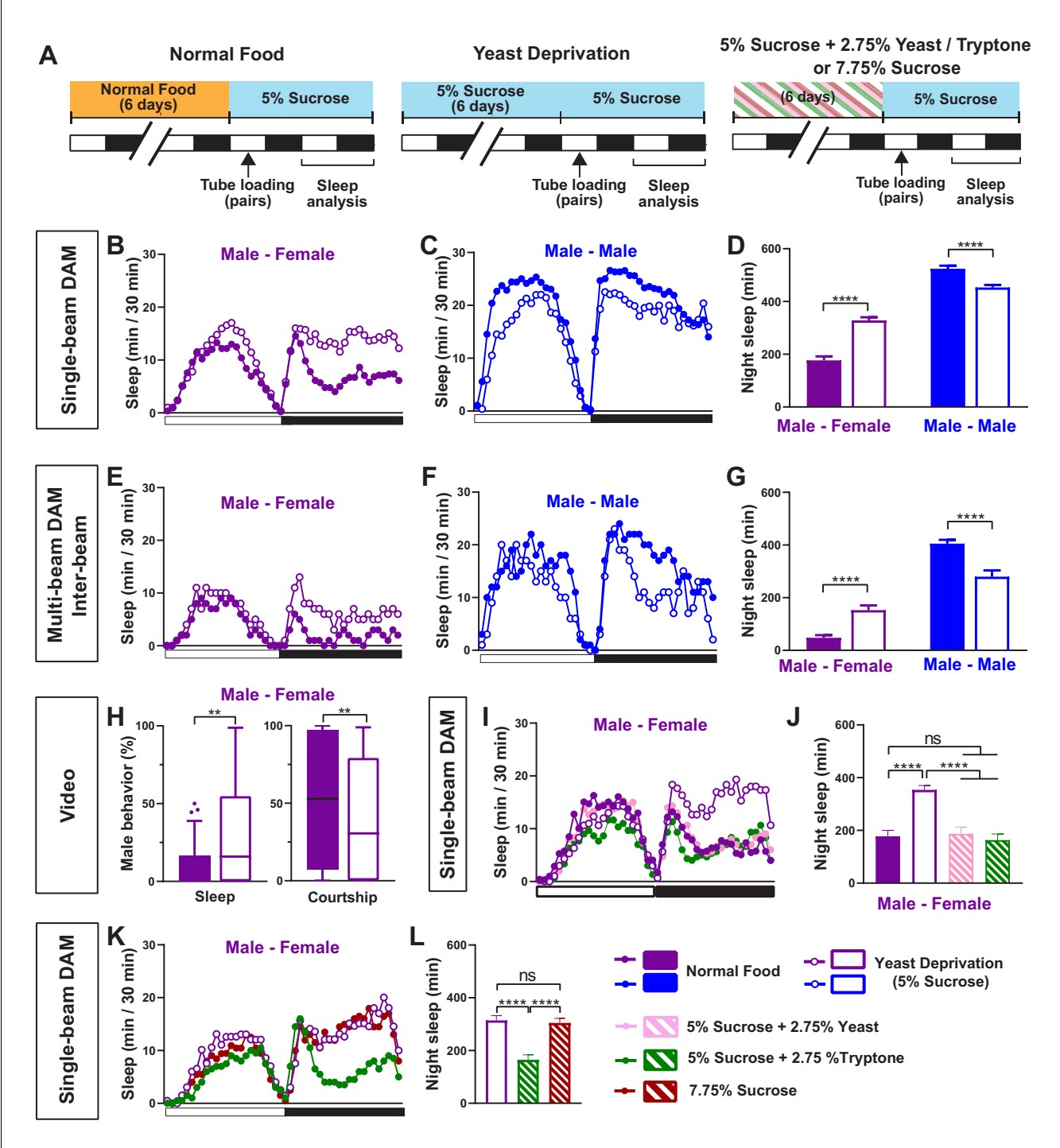

**Figure 1.** Yeast deprivation modulates the balance between sleep and courtship. (**A**) Schematic diagram of the experimental design. After 6 days in varying nutritional conditions as indicated, flies were loaded into tubes containing 5% sucrose in male-male (MM) or male-female (MF) pairs. (**B–C**) Sleep profile in 30 min intervals for MF (**B**) or MM (**C**) pairs in the normal food or yeast deprivation condition measured using single-beam monitors. N = 82–90. (**D**) Nighttime sleep for flies shown in (**B**) and (**C**). (**E–F**) Sleep profile in 30 min intervals for MF (**E**) or MM (**F**) pairs in the normal food or yeast deprivation condition measured using multi-beam monitors. Sleep was computed based on inter-beam movements, that is, movements between adjacent beams. N = 13–32. (**G**) Nighttime sleep for flies shown in (**E**) and (**F**). (**H**) Percent of time spent sleeping or courting for males in the normal food or yeast-deprived condition. Both groups of males were paired with normally fed females. The first 5 min of every h between Zeitgeber time (ZT) 18–24 were manually scored from videos. Tuckey's method is used for boxplots; boxes extend from the 25th to 75th percentiles, and whiskers extend from the lowest to the highest value within ±1.5 times the interquartile range. Data points above the whiskers are drawn as individual dots. N = 49–50.

*Figure 1 continued on next page*

*Figure 1 continued*

(I) Sleep profile of MF pairs in the normal food, 5% sucrose (yeast deprivation), 5% sucrose + 2.75% yeast, and 5% sucrose + 2.75% tryptone conditions. The single-beam DAM system was used to measure sleep. N = 45–48. (J) Nighttime sleep for flies shown in (I). (K) Sleep profile of MF pairs in 5% sucrose (yeast deprivation), 5% sucrose + 2.75% tryptone (tryptone supplemented), and 7.75% sucrose conditions. The tryptone supplemented and 7.75% sucrose conditions are equivalent in caloric content. The single-beam DAM system was used to measure sleep. N = 60–64. (L) Nighttime sleep for flies shown in (K). *Iso31* flies were used in all panels. In (B–D), MM and MF pairs were composed of flies from the same nutritional condition. In (E–L) and subsequent figures, 4- to 5-day-old normally fed males and females were used as partners for males from different nutritional conditions. In this and subsequent figures, bar graphs represent mean ± SEM and the white and black bars in the experimental design and below the x-axis in sleep profile graphs indicate light and dark periods, respectively. **p<0.01, ***p<0.001, ****p<0.0001 and ns: not significant, two-way ANOVA, p<0.0001 for the interaction between sex and nutritional condition, followed by Sidak post-hoc test (D, G); Mann-Whitney test (H); one-way ANOVA followed by Tukey post-hoc test (J, L).

The online version of this article includes the following figure supplement(s) for figure 1:

**Figure supplement 1.** Yeast deprivation does not promote sleep in individual flies.
**Figure supplement 2.** Yeast deprivation effects on sleep in MF pairs are independent of the nutritional condition of the female.
**Figure supplement 3.** Yeast deprivation modulates the balance between sleep and courtship: multi-beam data including local movements.

within a single beam such as feeding and grooming). The latter analysis can underestimate sleep because of twitches that occur during sleep (*Garbe et al., 2015*). As previously shown (*Garbe et al., 2015*), sleep measured with multi-beam monitors was lower than that measured with single-beam monitors, especially when both inter- and intra-beam movements (counts) were analyzed. Nevertheless, the effects of nutritional condition and social context on sleep were comparable between the two monitoring systems. As with single-beam data, multi-beam moves data showed that MF pairs with yeast-deprived males exhibited more nighttime sleep than normally fed MF pairs (*Figure 1E,G*), while yeast-deprived MM pairs slept less than normally fed MM pairs (*Figure 1F,G*). Even the highly sensitive counts analysis showed that MF pairs with yeast-deprived males slept significantly more than the normally fed counterparts during nighttime (*Figure 1—figure supplement 3*). Furthermore, video analysis of nighttime behavior confirmed that in MF pairs, yeast-deprived males slept significantly more than normally fed males (*Figure 1H*, *Video 1*). There was a corresponding reduction in the time spent performing courtship by yeast-deprived males compared with males that were kept in normal food until the behavioral assay (*Figure 1H*, *Video 1*). These results show that the single-beam DAM system can reliably measure the effects of nutritional status on sleep. Since the single-beam DAM system has higher throughput than the multi-beam DAM system or video analysis, we used the single-beam DAM system for quantifying sleep in subsequent experiments.

To determine whether the effects of the diet manipulation on MF sleep were attributable to yeast availability, we assayed the effects of adding yeast to the 5% sucrose diet and observed that the addition of 2.75% yeast was sufficient to restore the characteristic nighttime sleep suppression in MF pairs in the normal food condition (*Figure 1I,J*). Yeast contains both protein and lipids, and thus we next tested the effects of adding tryptone, a mixture of peptides generated by the tryptic digestion of casein, to the 5% sucrose diet. We found that the addition of 2.75% tryptone was sufficient to restore the normal nighttime sleep suppression in MF pairs (*Figure 1I,J*). Increasing the sucrose concentration to 7.75% to match the caloric content of 5% sucrose + 2.75% tryptone did not alter nighttime sleep in MF pairs compared to 5% sucrose (*Figure 1K,L*), suggesting that protein rather than caloric content is the important factor in regulating female-induced sleep loss. These data demonstrate that the balance between sleep and courtship in male flies is modulated by protein in dietary yeast.

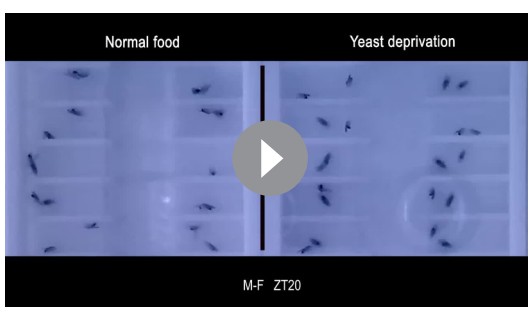

**Video 1.** Male-female pairs at ZT 20 under infrared light. Normally fed or yeast-deprived iso31 males were paired with normally fed female partners. While most normally fed males were courting, the majority of yeast-deprived males were inactive.
https://elifesciences.org/articles/60853#video1

## The effects of yeast deprivation on male sleep develop over multiple days but can be reversed within a few hours

In the experiments reported above, flies in the normal food condition were loaded into monitor tubes with 5% sucrose instead of normal food, because a pilot experiment suggested that food during sleep assay did not alter the effects of 6-day nutritional manipulations before the sleep assay. As a result, all flies were yeast deprived to some extent: either a little over a day (normal food condition) or a little over 7 days (yeast deprived condition) by the time of nighttime sleep assay. Therefore, our data suggest that yeast deprivation takes between 1 and 7 days to significantly impact male sleep. To examine the time course of the effects of yeast deprivation on sleep in more detail, we loaded normally fed males paired with females into monitor tubes containing either normal food or 5% sucrose food (*Figure 2A*). Statistically significant differences in nighttime sleep between the two conditions appeared on the third night, and the difference was more pronounced on the fourth night (*Figure 2B,C*), demonstrating that the effects of yeast deprivation on male sleep takes ~3 days to develop.

We next tested whether yeast-deprived males can quickly recover their characteristic female-induced sleep loss when normal diet is restored. To do so, males that were yeast deprived for several days were switched to normal food ~16 hr before the sleep assay (*Figure 2D*). MF pairs with yeast-deprived males that were switched to normal food exhibited sleep patterns similar to those of normally fed males (*Figure 2E–G*). These results demonstrate that the effects of yeast deprivation on male sleep can be reversed within a day of returning to a normal diet. To examine whether a shorter exposure to normal food would suffice to reverse the effects of yeast deprivation, we first loaded yeast-deprived males paired with females into tubes containing 5% sucrose. We then transferred the MF pairs to new tubes containing either normal food or 5% sucrose (to control for the effects of handling) within the last 20 min of the light period and assayed nighttime sleep following the transfer (*Figure 2H*). Flies transferred to normal food exhibited significantly reduced sleep compared to flies transferred to sucrose food, which becomes apparent within 2 hr of the transfer (*Figure 2I,J*). Together, our data demonstrate that yeast deprivation takes multiple days to alter male sleep patterns, but its effects can be reversed within 2 hr of normal feeding.

## Yeast deprivation does not impair the ability to perform reproductive behaviors in males

Males may prioritize sleep over courtship under non-optimal nutritional conditions. Under a normal nutritional condition, males forgo sleep to engage in courtship at night. However, nutritional restriction, which likely results in unfavorable reproductive outcomes, may tip the balance toward sleep. Alternatively, male flies may have difficulty courting and mating after an extended period of yeast deprivation. To distinguish between these possibilities, we performed courtship assays between ZT 1–4, when flies are generally awake. We paired virgin males and virgin females and measured the courtship index and latency under normally fed and yeast deprived conditions. Interestingly, no difference in courtship index or latency was found between the two nutritional conditions (*Figure 3A–C*). Similarly, yeast-deprived males were as successful at copulation as their normally fed counterparts (*Figure 3D*). These data show that the ability to perform reproductive behaviors is not impaired by several days of yeast deprivation and suggest that nutritional conditions modulate the balance between sleep and courtship rather than courtship per se.

## Dopaminergic neurons projecting to the protocerebral bridge act downstream of male-specific P1 neurons

Male-specific P1 neurons are primarily known for the control of courtship behaviors (*Kohatsu et al., 2011*; *Pan et al., 2012*; *von Philipsborn et al., 2011*). However, our previous work and work from other groups have established that activation of P1 neurons leads to sleep suppression, suggesting that they also play a role in regulating sleep (*Beckwith et al., 2017*; *Chen et al., 2017*; *Machado et al., 2017*). Since yeast deprivation impacts the balance between sleep and courtship, we examined whether the sleep-suppressive effects of P1 activation are modulated by nutrition. As observed previously, activation of P1 neurons using the P1 split Gal4 driver (*Inagaki et al., 2014*) and the warmth-sensitive TrpA1 channel (*Hamada et al., 2008*) resulted in decreased sleep in normally fed males (*Figure 4A–C*). Interestingly, the sleep suppressing effects of P1 activation was

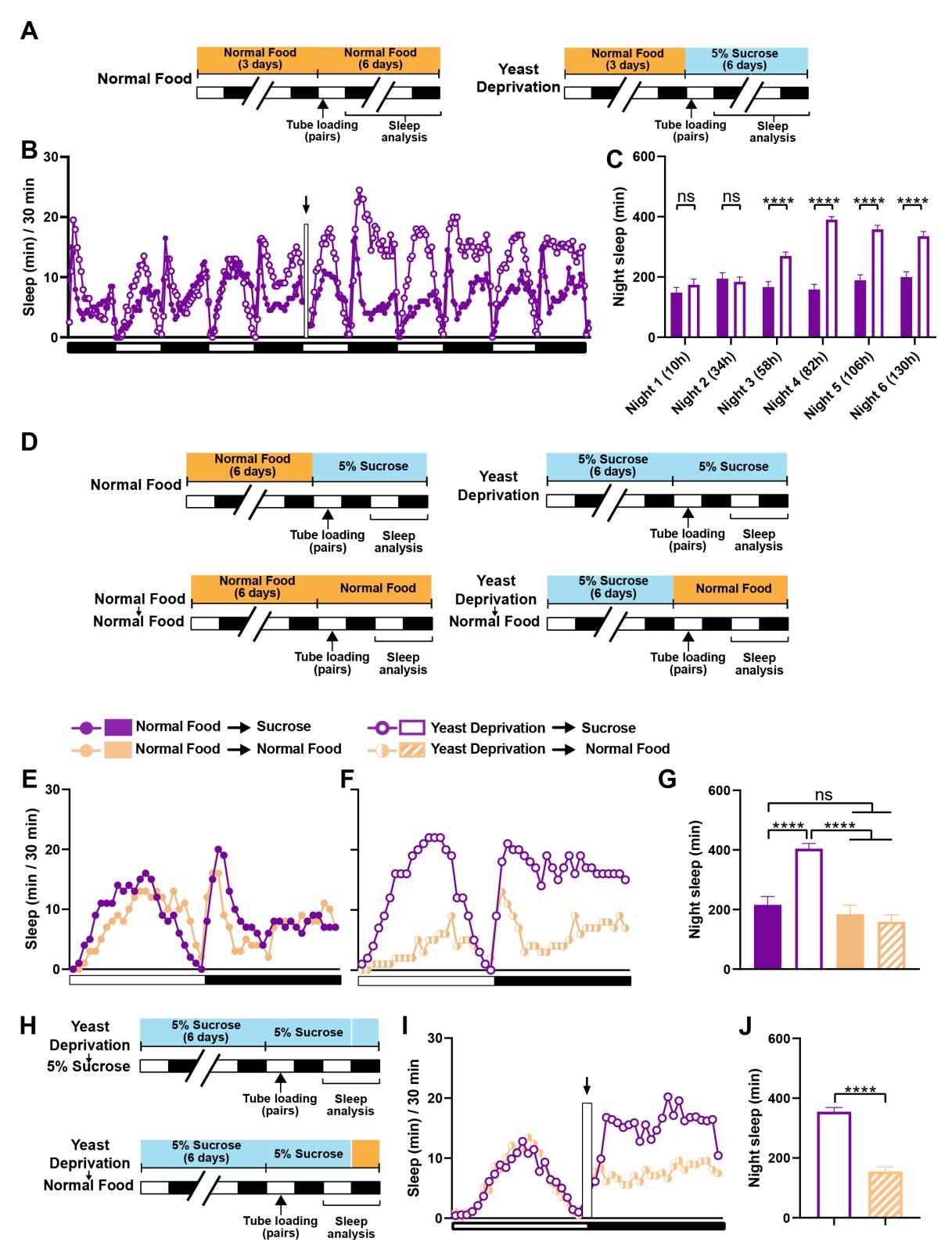

**Figure 2.** The effects of yeast deprivation on male sleep develop over multiple days but can be reversed within a few hours. (**A**) Schematic diagram of the experimental design for (**B**) and (**C**). Normally fed MF pairs were loaded into tubes containing 5% sucrose or normal food between ZT 1–2, and sleep was assayed for six nights, starting at ZT 12 on the loading day. (**B**) Sleep profile in 30 min intervals for normally fed MF pairs loaded into tubes containing either 5% sucrose food or normal food. Flies were transferred to new tubes after 3 days to circumvent the potential problem of larval

*Figure 2 continued on next page*

*Figure 2 continued*

movements interfering with sleep measurements. The rectangle and arrow indicate the time of transfer. N = 59–60. (C) Nighttime sleep for the flies shown in (B) Time from loading until the begining of each night is indicated in parenthesis. (D) Schematic diagram of the experimental design for (E), (F) and (G). After 6 days in the indicated nutritional conditions, MF pairs were loaded into tubes containing either 5% sucrose or normal food. (E) Sleep profile in 30 min intervals for MF pairs in the normal food condition loaded into tubes containing either 5% sucrose food or normal food. (F) Sleep profile in 30 min intervals for MF pairs in the yeast deprivation condition loaded into tubes containing either 5% sucrose food or normal food. N = 31–32. (G) Nighttime sleep for the flies shown in (E) and (F). (H) Schematic diagram of the experimental design for (I) and (J). MF pairs yeast-deprived for 6 days were loaded into tubes containing 5% sucrose. Flies were transferred to tubes containing either 5% sucrose or normal food at ZT 12 on the following day. (I) Sleep profile in 30 min intervals for yeast-deprived MF pairs transferred into tubes containing either 5% sucrose food or normal food. The rectangle and arrow indicate the time of transfer. N = 51–54. (J) Nighttime sleep for the flies shown in (I). *Iso31* flies were used in all panels. ****p<0.0001 and ns: not significant, two-way ANOVA followed by Sidak post hoc test (C), one-way ANOVA followed by Tukey post hoc test (G), unpaired t-test (J).

absent (for daytime sleep) or reduced (for nighttime sleep) in male flies that were yeast deprived for 8 days (6 days before loading and 2 days after loading) prior to activation (*Figure 4B,C*). This suggests that P1 neurons or the circuit downstream of P1 neurons are modulated by dietary yeast.

Previous studies have identified several neuronal clusters that act downstream of P1 neurons to regulate courtship (*Kohatsu et al., 2011*; *von Philipsborn et al., 2011*). However, little is known about the sleep circuit downstream of P1 neurons. DN1 clock neurons have been shown to act both upstream and downstream of P1 neurons for sleep regulation, but the DN1-P1 connections appear to be indirect (*Chen et al., 2017*). To identify candidate neurons acting directly downstream of P1 for sleep regulation, we employed the trans-Tango trans-synaptic tracing technique (*Talay et al.,*

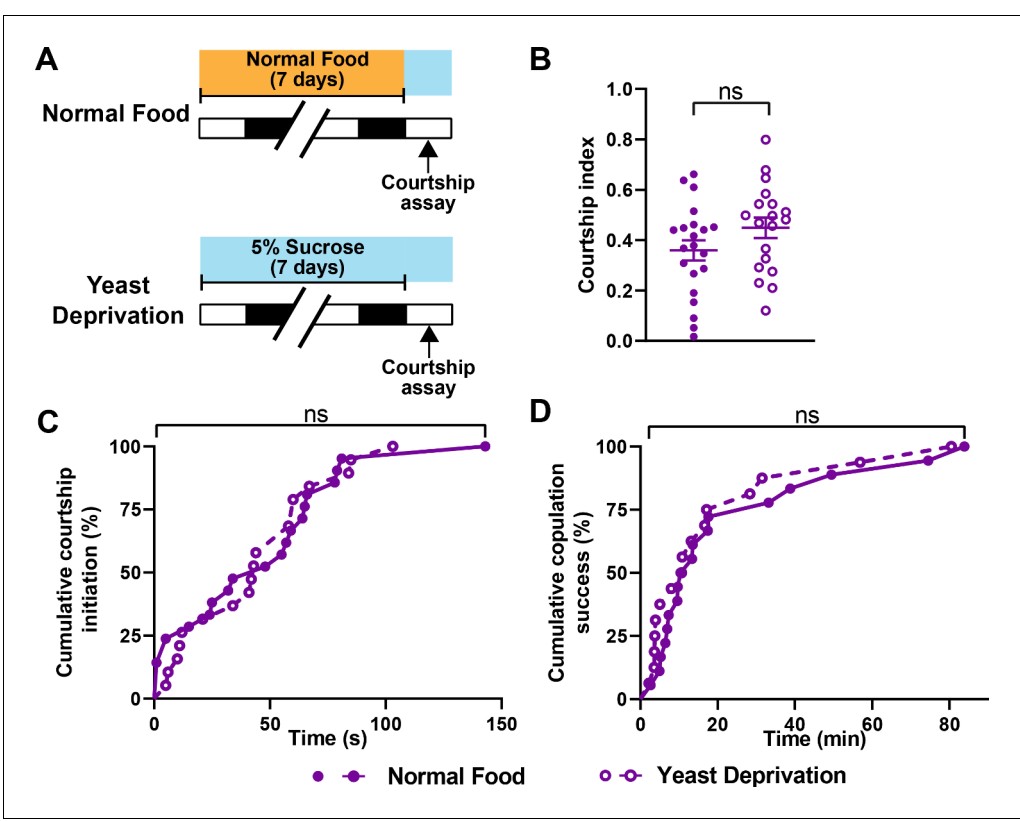

**Figure 3.** Yeast deprivation does not impair the ability of male flies to perform reproductive behaviors. (A) Schematic diagram of the experimental design. After 7 days in indicated nutritional conditions, males were paired with virgin females in an arena containing 5% sucrose. (B–D) Courtship index (B), latency to court (C), and latency to copulation (D) in males in the normal food or yeast deprivation condition. N = 19–21. Courtship/mating assay was performed between ZT 1–4. *Iso31* flies were used in all panels. ns: not significant, unpaired t-test (B); log-rank test (C, D).

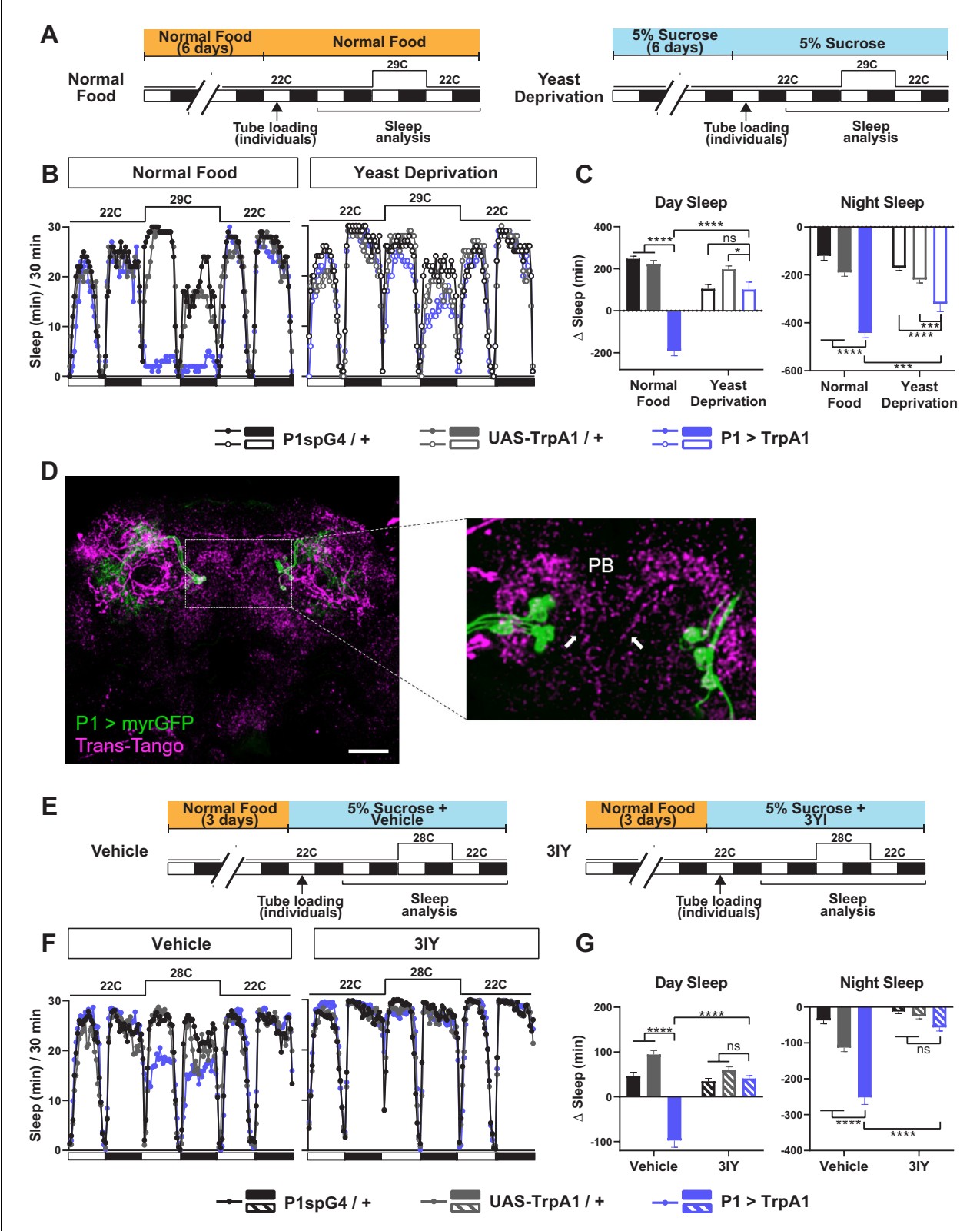

**Figure 4.** Yeast deprivation and inhibition of dopamine signaling impairs the wake-promoting effects of P1 activation, and trans-Tango tracing identifies potential downstream targets of P1 neurons. (A) Schematic diagram of the experimental design for (B) and (C). (B) Sleep profiles in 30 min intervals for experimental (P1 >TrpA1) and parental control (P1-spG4 / + and UAS-TrpA1 / +) males in normal food or yeast deprivation conditions, loaded into tubes containing normal food or 5% sucrose, respectively. N = 29–32. TrpA1 was activated by raising the temperature from 22°C to 29°C. *Figure 4 continued on next page*

*Figure 4 continued*

(**C**) Daytime and nighttime sleep change (sleep at 29˚C – baseline sleep at 22˚C) for flies shown in (**B**). (**D**) Confocal projection of an adult male brain in which trans-Tango was driven by P1 split Gal4 (spG4). Presynaptic P1 neurons express myrGFP (green) and postsynaptic targets express mtdTomato (red). Right image shows a magnification of the PB region, with postsynaptic neurons that innervate the PB. Arrows indicate descending projections used to identify DA-PB neurons. Scale bar represents 50 µm. (**E**) Sleep profiles in 30 min intervals for experimental (P1 >TrpA1) and parental control (P1-spG4 / + and UAS-TrpA1 / +) male flies. Flies were raised on normal food and individually loaded into tubes containing 5% sucrose supplemented with vehicle (propionic acid) or 3IY (inhibitor of dopamine synthesis). N = 40–48. TrpA1 was activated by raising the temperature from 22˚C to 28˚C. (**G**) Daytime and nighttime sleep change (sleep at 28˚C – baseline sleep at 22˚C) for flies shown in (**F**). ***p<0.001, ****p<0.0001, ns: not significant, two-way ANOVA, followed by Tukey post-hoc test (**C**) and (**G**); p<0.0001 for the interaction between genotype and nutritional condition (**C**); p<0.0001 for the interaction between genotype and drug condition (**G**).

*2017*). By introducing genetically engineered ligand-receptor pairs, trans-Tango allows induction of mtdTomato expression in postsynaptic partners of defined presynaptic neurons. Several neuronal clusters were revealed by trans-Tango as potential postsynaptic partners of the P1 cluster (*Figure 4D*). They included PB-projecting neurons with descending projections in the midline.

We selected the PB-projecting neurons for further investigation for several reasons. First, the PB is a compartment of the central complex, a set of neuropils in the center of the brain, and has extensive connections with the other central complex compartments: the fan-shaped body (FB), ellipsoid body (EB), and noduli (NO). Notably, several groups of central complex neurons projecting to the FB and EB have been implicated in sleep regulation (*Donlea et al., 2014*; *Donlea et al., 2011*; *Liu et al., 2012*; *Liu et al., 2016*; *Pimentel et al., 2016*; *Ueno et al., 2012*). Thus, it seemed likely that the PB would also be involved in sleep regulation. Second, the entire population of PB-projecting neurons have been extensively characterized (*Hanesch et al., 1989*; *Lin et al., 2013*; *Young and Armstrong, 2010*), and only a single pair of PB-projecting neurons, named PB.b-LAL.s-PS.s, or LPsP, have descending projections in the midline (*Wolff et al., 2015*; *Wolff and Rubin, 2018*). And lastly, the LPsP neurons correspond to the only pair of dopaminergic neurons that project to the PB, previously named T1 (*Alekseyenko et al., 2013*; *Nässel and Elekes, 1992*). This is particularly interesting because we found that sleep suppression by P1 activation requires dopaminergic signaling. When dopamine synthesis was inhibited by 3-Iodo-L-tyrosine (3IY), activation of P1 neurons did not suppress sleep in males (*Figure 4E–G*). Based on these considerations, we investigated whether the dopaminergic PB-projecting neurons, which we will refer to as DA-PB neurons, act downstream of P1 to balance sleep and courtship.

To confirm the anatomical connection between P1 and DA-PB neurons, we first determined whether P1 neurons send axonal projections to the PB. We expressed the presynaptic protein Synaptotagmin (Syt) fused with GFP in P1 neurons, and found a clear presence of Syt::GFP-marked presynaptic sites in the PB region, although the signal in this structure was not as strong as in other brain regions (*Figure 5A*). Next, we asked whether DA-PB neurons have dendrites in the PB. Previous morphological analysis suggested that DA-PB neurons contain both presynaptic and postsynaptic connections in the PB region (*Wolff et al., 2015*). To confirm this, we simultaneously expressed the postsynaptic marker DenMark (*Nicolaï et al., 2010*) and presynaptic GFP-tagged Syt protein (*Zhang et al., 2002*) in DA-PB neurons using a specific Split-Gal4 driver line (SS52578, *Wolff and Rubin, 2018*). We found that the postsynaptic DenMark signal was present in the PB region (*Figure 5B*, left image), confirming that DA-PB neurons are in a position to receive inputs from P1 neurons. The DenMark signal was also present in the lateral accessory lobe (LAL), and the presynaptic Syt::GFP signal was found mainly in the PB region (*Figure 5B*). Similar patterns of pre- and postsynaptic markers were observed in females, suggesting that these neurons are not sexually dimorphic at the gross morphological level (*Figure 5—figure supplement 1*). These data, in combination with the trans-Tango data, suggest that DA-PB neurons are direct downstream partners of P1 neurons.

We next examined whether the P1 cluster and DA-PB neurons are functionally connected. We expressed the ATP-sensitive P2X2 receptor (*Lima and Miesenböck, 2005*) in P1 neurons and the calcium sensor GCaMP6m (*Chen et al., 2013*) in DA-PB neurons. We found that activation of P1 cells using 2.5 mM ATP perfusion led to a marked increase in GCaMP6m signal in the PB region of DA-PB neurons, compared with controls flies which did not express P2X2 (*Figure 5C*), pointing to an

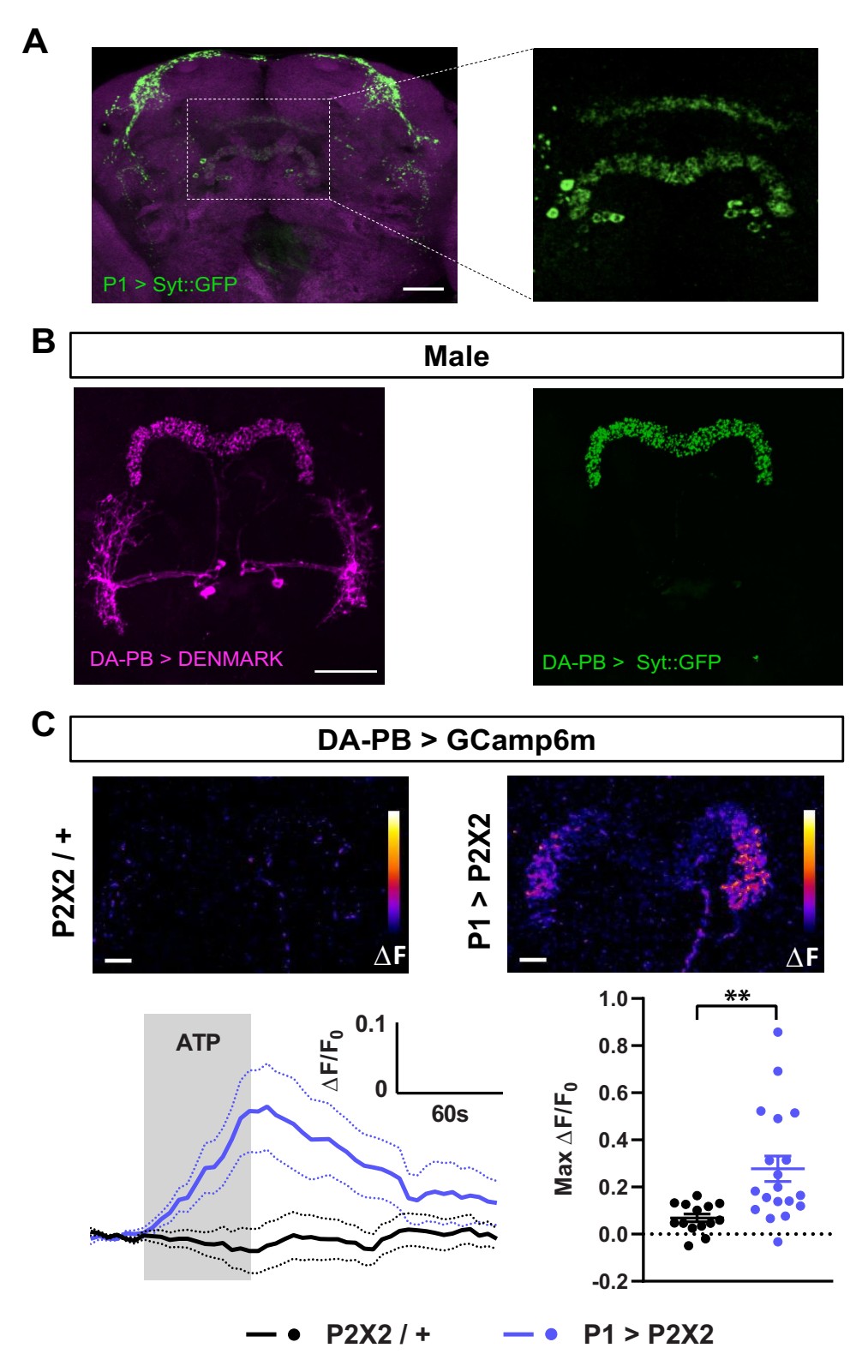

**Figure 5.** Dopaminergic neurons projecting to the protocerebral bridge (DA-PB) act downstream of male-specific P1 neurons. (**A**) Confocal projection of an adult male brain in which Syt::GFP was driven by P1-spG4. Anti-Bruchpilot (BRP, magenta) was used to localize neuropil regions. The image on the right shows a magnification of the PB region, which contains presynaptic terminals from P1 neurons. (**B**) Confocal projection of an adult male brain in which DenMark (postsynaptic marker, left) and Syt::GFP (presynaptic marker, right) are expressed in DA-PB neurons. Both postsynaptic and

*Figure 5 continued on next page*

*Figure 5 continued*

presynaptic makers are expressed in the PB region. (C) Increase in GCaMP6m signal (ΔF) in the PB projections of DA-PB neurons upon perfusion with ATP of a male brain expressing P2X2 in P1 neurons (P1 >P2X2, top right) or a genetic control (P2X2/+, top left). Fluorescence traces (bottom left) and peak responses (bottom right) for normalized GCaMP6m response ($\Delta F/F_0$) in the PB projections of DA-PB neurons in response to P1 activation (blue, P1 >P2X2) compared with the genetic control (black, P2X2/+). R71G01-lexA was used to express P2X2 in P1 neurons and SS52578 spG4 was used to express GCaMP6m in DA-PB neurons. Grey rectangle indicates 2.5 mM ATP perfusion. N = 14–19. Scale bars represent 50 µm in (A–B) and 10 µm in (C). **$p<0.01$, unpaired t-test with Welch's correction for unequal variances.

The online version of this article includes the following figure supplement(s) for figure 5:

**Figure supplement 1.** Female DA-PB neurons show similar projection patterns as male counterparts.

excitatory connection between P1 and DA-PB neurons. Together, these data demonstrate that DA-PB neurons are anatomically and functionally downstream of P1 neurons.

## DA-PB neurons regulate male sleep in a nutrition-dependent manner

Since DA-PB neurons act downstream of the sleep-suppressing P1 cluster, DA-PB neurons may also be involved in sleep regulation. To test the sleep-regulatory role of DA-PB neurons in both yeast-deprived and normally fed flies, we expressed TrpA1 channel in DA-PB neurons and activated them by increasing the ambient temperature (*Figure 6A*). Normally fed males with activated DA-PB neurons showed small, but significant sleep suppression during the nighttime relative to control males (*Figure 6B,D*). Notably, the sleep-suppressing effects of DA-PB activation were not detectable in yeast-deprived males (*Figure 6C,D*), suggesting that the impact of DA-PB activation depends on the nutritional conditions. Although normally fed females with activated DA-PB neurons showed significant differences in sleep compared to both parental controls (*Figure 6E,G*), the differences were in opposite directions, and yeast-deprived females with activated DA-PB neurons exhibited a similar amount of sleep as one of the parental controls (*Figure 6F,G*). These results do not support the role of DA-PB neurons in female sleep. Together, our data show that DA-PB neurons are involved in nutrition-dependent sleep regulation in males.

## A screen identifies an additional PB-projecting neuronal group that regulates sleep

To map neurons downstream of DA-PB neurons, we performed trans-Tango trans-synaptic tracing experiments (*Talay et al., 2017*). We found that neurons that arborize in other central complex compartments, the EB, FB, and NO, are the major postsynaptic partners of DA-PB neurons (*Figure 7A*). To determine the identity of specific neuronal groups acting downstream of DA-PB neurons, we conducted a screen of PB-projecting neuronal groups. Since we found that activation of DA-PB neurons suppresses sleep in normally fed males but not in females, we examined sleep in both males and females under the normal food conditions. We activated various PB-projecting neuronal groups using previously characterized split-Gal4 lines (*Wolff and Rubin, 2018*) and UAS-TrpA1. The screen identified two candidate neuronal groups that regulate sleep in males (*Figure 7B*): P-EG neurons (SS02198) projecting from the PB to the EB and gall (*Figure 7C*; *Wolff and Rubin, 2018*) and P-FN$_{m-p}$ neurons (SS52244) projecting from the PB to the ventral FB and medial and posterior NO3 (*Figure 7—figure supplement 1A*). Further experiments confirmed that activation of P-EG neurons leads to nighttime sleep suppression in males, but not in females (*Figure 7D,E*, *Figure 7—figure supplement 1A-C*). However, we could not confirm sleep suppression by P-FN$_{m-p}$ activation, suggesting that these neurons play a minor role in sleep regulation, if any (*Figure 7—figure supplement 1D–F*). Overall, our data suggest that P-EG neurons interact with DA-PB neurons and regulate sleep in a sex-dependent manner.

## Discussion

The integration of environmental cues and internal states is critical for selecting behaviors that optimize animals' evolutionary fitness under varying conditions. Whereas sleep is thought to be regulated mainly by the circadian and homeostatic processes (*Borbély, 1982*), other motivational factors play critical roles in modulating sleep. Competition between sleep and other needs is a general phenomenon documented in many species. Examples include sleep suppression in migrating birds

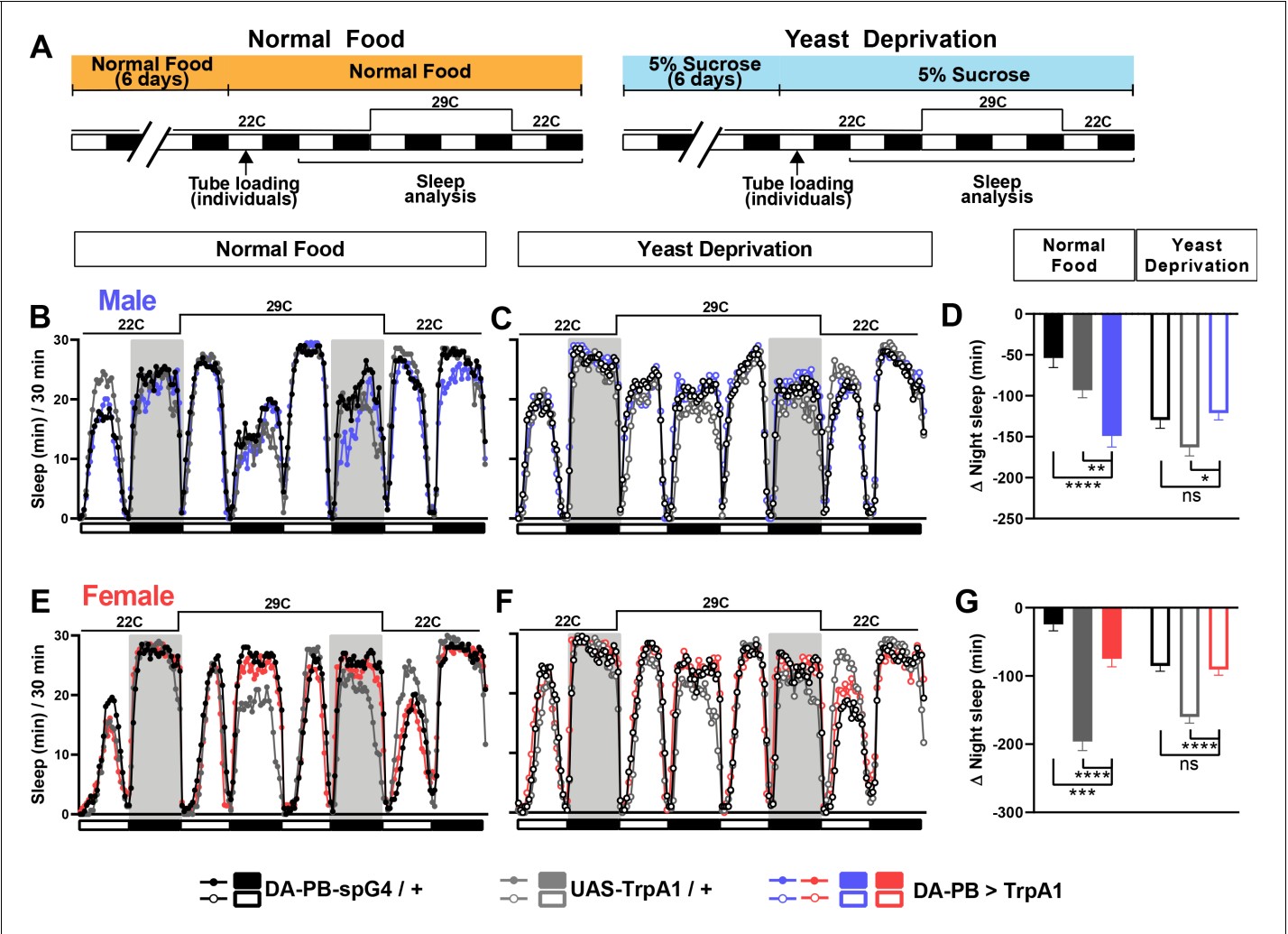

**Figure 6.** DA-PB neurons regulate male sleep in a nutrition-dependent manner. (A) Schematic diagram of the experimental design. After 6 days in vials containing normal food or 5% sucrose food, male and female flies were loaded individually into tubes. (B–C) Sleep profiles in 30 min intervals for experimental (DA-PB >TrpA1) and parental control (DA-PB-spG4 / + and UAS-TrpA1 / +) male flies in the normal food (B) or yeast deprivation (C) condition. N = 59–62. TrpA1 was activated by raising the temperature from 22°C to 29°C. (D) Nighttime sleep change (sleep during the second night at 29°C – baseline night sleep at 22°C) for the flies shown in (B) and (C). (E–F). Sleep profiles in 30 min intervals of experimental (DA-PB >TrpA1) and parental control (DA-PB-spG4 / + and UAS-TrpA1 / +) female flies in the normal food (E) and yeast deprivation (F) condition. N = 48–56. (G) Nighttime sleep change for the flies shown in (E) and (F). ***p<0.001, ****p<0.0001 and ns: not significant, two-way ANOVA, p<0.0001 for the interaction between genotype and nutritional condition, followed by Tukey post-hoc test (C–F).

(*Rattenborg et al., 2016*; *Rattenborg et al., 2004*), in male arctic sandpipers during an annual 3 week mating season (*Lesku et al., 2012*), and in flies and worms during unexpected starvation (*Goetting et al., 2018*; *Keene et al., 2010*). Another example are the Mexican cavefish, who live in an environment with limited and seasonal food availability. They show increased sleep upon starvation, suggesting that they sleep more during the dry season of food scarcity to conserve energy and sleep less during the wet season of relative food abundance to forage (*Jaggard et al., 2017*). In addition, we and others have previously shown that when presented with a female partner, *Drosophila* males forgo nighttime sleep to engage in courtship (*Beckwith et al., 2017*; *Machado et al., 2017*). *Beckwith et al., 2017* further showed that male flies do not exhibit rebound sleep after prolonged wakefulness in the presence of females and that female pheromone can suppress male rebound sleep after sleep deprivation by mechanical stimulation. These results suggest that male sexual arousal can inhibit sleep even when sleep drive is high. We also showed previously that

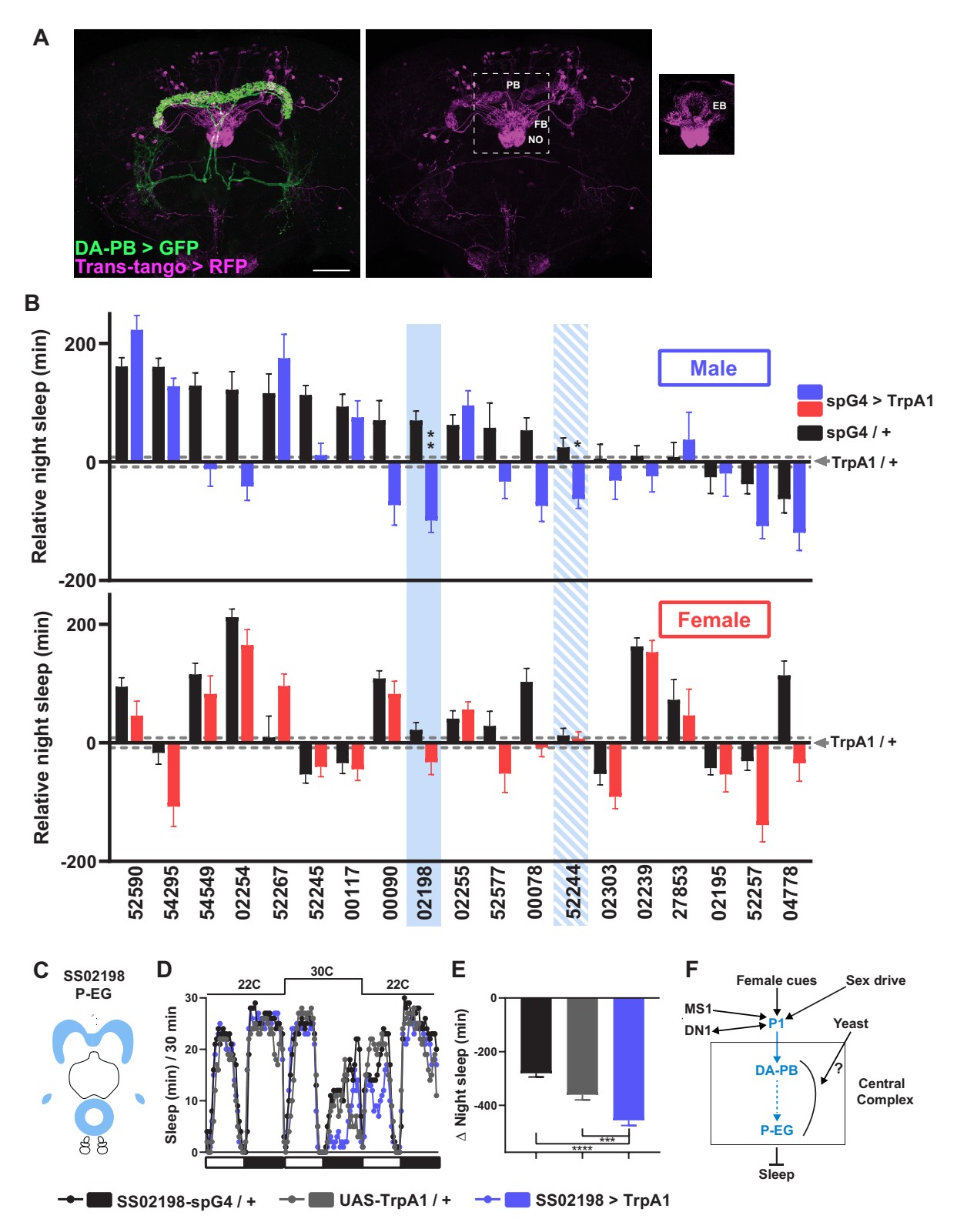

**Figure 7.** A screen identifies additional PB-projecting neurons that regulate sleep. (**A**) Confocal projection of an adult male brain in which trans-Tango was driven by the SS52578 DA-PB spG4 driver. Presynaptic DA-PB neurons express MyrGFP (green) and postsynaptic targets express mtdTomato (red). Postsynaptic targets were detected in the PB, FB, and NO (middle panel) as well as in the EB (right panel; since expression in the EB was masked in the full projection, only two image slices were included in the projection). Scale bar: 50 μm. (**B**) Nighttime sleep in individual, normally fed males (top) and

*Figure 7 continued on next page*

*Figure 7 continued*

females (bottom) in response to activation of various neuronal groups projecting to the PB. TrpA1 was activated by raising the temperature from 22°C to 30°C. Labels on the X-axis refer to the identity of the SS spG4 lines from the Rubin lab spG4 collection. The difference in nighttime sleep for each spG4 line with respect to the UAS-TrpA1 control is plotted. Blue and red bars represent male and female experimental groups (spG4 >TrpA1), respectively, and black bars represent the Gal4 control (spG4 / +). Dashed grey lines indicate the SEM of the UAS-TrpA1 / + control. N = 144–148 for UAS-TrpA1 / + controls, and N = 15–52 for experimental groups and Gal4 / + controls. Blue rectangles highlight spG4 lines associated with significant sleep changes in males relative to both controls. The solid rectangle represents a spG4 line confirmed in a re-test, whereas the striped rectangle represents a line not confirmed in a re-test. None of the sleep changes in females were significant relative to both controls. For simplicity, non-significant differences are not indicated. (**C**) Schematic diagrams of the expression pattern of the SS02198 spG4 driver (based on *Wolff and Rubin, 2018*). (**D**) Sleep profiles in 30 min intervals for experimental (SS02198 >TrpA1) and parental control (SS02198-spG4 / + and UAS-TrpA1 / +) males. Normally fed flies were loaded into tubes with 5% sucrose. TrpA1 was activated by raising the temperature from 22°C to 30°C. N = 27–32. (**E**) Nighttime sleep change (sleep at 30°C – baseline sleep at 22°C) for the flies shown in (**D**). *p<0.05, **p<0.01, ***p<0.001, Brown-Forsythe and Welch ANOVA for unequal variances followed by Dunnett T3 post hoc test (**B**), one-way ANOVA followed by Dunnet post hoc test (**D**). (**F**) A working model of the neural mechanisms integrating sleep drive, sex drive, and yeast hunger in *Drosophila* males. Previous studies have shown that P1 neurons integrate female and male pheromonal cues from multiple Fru$^{M}$-expressing neuronal clusters (*Kohatsu et al., 2011*; *Stockinger et al., 2005*), octopaminergic arousal signal from MS1 neurons (*Machado et al., 2017*), and circadian information from DN1 clock neurons (*Chen et al., 2017*). Current work suggests that DA-PB neurons act downstream of P1 neurons and upstream of P-EG neurons to promote wakefulness in a male-specific and nutrition-dependent manner.

The online version of this article includes the following figure supplement(s) for figure 7:

**Figure supplement 1.** P-EG activation does not affect female sleep, and P-FN$_{m-p}$ activation has little effect on sleep in males and females.

increased sleep drive (due to sleep deprivation) or reduced sex drive (due to recent copulations) tilts the sleep-courtship balance toward more sleep and less courtship during the night (*Machado et al., 2017*). Our present results show that yeast deprivation also tilts the balance toward more nighttime sleep.

Interestingly, yeast deprivation has little effect on daytime courtship in our study. Previous research on the effects of dietary yeast on male reproductive fitness found variable results depending on the experimental design. Yeast content had little or a non-monotonic influence on the number of offspring when males competed with other males (*Fricke et al., 2008*). On the other hand, the amount of dietary yeast was negatively correlated with the number of offspring when no male-male competition was involved (*Zajitschek et al., 2013*). However, these studies did not include a condition where yeast was absent, and thus do not provide insights into yeast deprivation's effects on male sexual performance. Our current data suggest that dietary yeast influences the male fly's willingness to stay awake to engage in courtship at night, but does not impair their ability to court during the day when they are usually awake. Since yeast provides essential nutrients for larval development (*Becher et al., 2012*), our findings suggest that flies engage in a sophisticated cost-benefit analysis that takes nutritional status into account in deciding whether the potential benefit of pursuing female partners is worth the cost of losing sleep.

Dietary yeast is the primary source of protein and lipids in the standard laboratory food for flies. Our finding that tryptone can substitute for yeast demonstrates that the effects of yeast deprivation are primarily due to the lack of protein. Previous studies have identified several neuronal populations that mediate the effects of dietary protein and amino acids on adult *Drosophila* behavior. These include dopaminergic Wedge neurons, EB Ring5 neurons, and peptidergic neurons expressing diuretic hormone-44, insulin-like peptide-2, or leucokinin (*Brown et al., 2020*; *Ki and Lim, 2019*; *Liu et al., 2017*; *Murphy et al., 2016*; *Yang et al., 2018*; *Yurgel et al., 2019*). It would be interesting to determine whether these neurons are involved in modulating the sleep-courtship balance by nutrition. It is noteworthy that DA-PB neurons have been shown to regulate of male aggression (*Alekseyenko et al., 2013*). Male flies engage in aggressive behavior to compete for resources such as food and female partners (*Lim et al., 2014*; *Yuan et al., 2014*). DA-PB neurons may be involved in integrating pheromonal cues and nutritional status to regulate the balance between sleep and aggression or courtship, depending on the context.

A number of neuronal populations that regulate sleep have been identified (*Artiushin and Sehgal, 2017*; *Tomita et al., 2017*), and among them are two distinct populations in the central complex: EB R5 neurons and the dorsal FB (*Donlea et al., 2014*; *Donlea et al., 2011*; *Liu et al., 2012*; *Liu et al., 2016*; *Pimentel et al., 2016*; *Ueno et al., 2012*). In addition, *Ueno et al., 2012* showed

that activation of dopaminergic PPM3 neurons projecting to the ventral FB leads to sleep suppression, while *Dag et al., 2019* showed that ventral FB neurons can be sleep-promoting. Our results show that the PB region in the central complex is also involved in sleep regulation. Activation of DA-PB neurons, as well as P-EG neurons acting downstream of them to convey information from the PB to the EB, suppress sleep in males. It would be interesting to determine whether P-EG neurons interact with the previously described neurons projecting to the EB. Based on the present and previous data, we propose that P1, DA-PB, and P-EG neurons, as well as previously described octopaminergic MS1 neurons and DN1 clock neurons (*Chen et al., 2017*; *Machado et al., 2017*), form a male-specific sleep circuit (*Figure 7F*). Our finding that DA-PB activation leads to sleep suppression in normally fed, but not yeast-deprived, males suggests that information about yeast availability is conveyed to the male sleep circuit at the level of DA-PB neurons or downstream of them. The information could be transmitted in the form of inhibitory inputs from neurons encoding yeast hunger or excitatory inputs from neurons encoding yeast satiety. Further research would be required to determine how information about yeast availability is integrated into the circuit.

Sleep is strongly influenced by monoaminergic neuromodulators, including dopamine, serotonin, and octopamine and its mammalian analog norepinephrine (*Griffith, 2013*; *Joiner, 2016*; *Liu et al., 2019*; *Nall and Sehgal, 2014*; *Ni et al., 2019*; *Singh et al., 2015*). We previously showed that octopamine is a significant mediator of sleep suppression by male sex drive upstream of P1 neurons (*Machado et al., 2017*). Our present data show that dopamine signaling functions downstream of P1 neurons in the process. This is reminiscent of several studies showing that octopamine/norepinephrine acts upstream of dopaminergic neurons for diverse biological processes including memory, feeding, and addiction (*Burke et al., 2012*; *Goertz et al., 2015*; *Wang et al., 2016*). Octopamine/norepinephrine may provide an arousal signal that enhances dopaminergic control of motivated behaviors. A similarly layered signaling may underlie the integration of sleep and other motivated behaviors in flies and mammals.

# Materials and methods

## Key resources table

| Reagent type (species) or resource | Designation | Source or reference | Identifiers | Additional information |
|---|---|---|---|---|
| Genetic reagent (*D. melanogaster*) | GMR71G01-*LexA* | Bloomington *Drosophila* Stock Center | BDSC #54733 | |
| Genetic reagent (*D. melanogaster*) | UAS-*TrpA1* | Bloomington *Drosophila* Stock Center | BDSC #26263 | |
| Genetic reagent (*D. melanogaster*) | lexAop-*P2X2* | Bloomington *Drosophila* Stock Center | BDSC #76030 | |
| Genetic reagent (*D. melanogaster*) | UAS-*GCaMP6m* | Bloomington *Drosophila* Stock Center | BDSC #42750 | |
| Genetic reagent (*D. melanogaster*) | UAS-*Syt::GFP* | Bloomington *Drosophila* Stock Center | BDSC #6925 | |
| Genetic reagent (*D. melanogaster*) | UAS-*Denmark* | Bloomington *Drosophila* Stock Center | BDSC #33061 | |
| Genetic reagent (*D. melanogaster*) | *trans-Tango*; UAS-*myrGFP*, QUAS-*mtd Tomato-3xHA* | Bloomington *Drosophila* Stock Center | BDSC #77124 | |
| Genetic reagent (*D. melanogaster*) | Iso31 (*w*[1118]) | Bloomington *Drosophila* Stock Center | BDSC #3605 | |

*Continued on next page*

*Continued*

| Reagent type (species) or resource | Designation | Source or reference | Identifiers | Additional information |
|---|---|---|---|---|
| Genetic reagent (*D. melanogaster*) | P1-split Gal4 | David Anderson | | *Inagaki et al., 2014* |
| Genetic reagent (*D. melanogaster*) | PB Split-Gal4 line, SS52578 | Janelia Research Campus | | *Wolff and Rubin, 2018* |
| Genetic reagent (*D. melanogaster*) | PB Split-Gal4 line, SS54295 | Janelia Research Campus | | *Wolff and Rubin, 2018* |
| Genetic reagent (*D. melanogaster*) | PB Split-Gal4 line, SS52590 | Janelia Research Campus | | *Wolff and Rubin, 2018* |
| Genetic reagent (*D. melanogaster*) | PB Split-Gal4 line, SS52245 | Janelia Research Campus | | *Wolff and Rubin, 2018* |
| Genetic reagent (*D. melanogaster*) | PB Split-Gal4 line, SS02254 | Janelia Research Campus | | *Wolff and Rubin, 2018* |
| Genetic reagent (*D. melanogaster*) | PB Split-Gal4 line, SS52267 | Janelia Research Campus | | *Wolff and Rubin, 2018* |
| Genetic reagent (*D. melanogaster*) | PB Split-Gal4 line, SS02255 | Janelia Research Campus | | *Wolff and Rubin, 2018* |
| Genetic reagent (*D. melanogaster*) | PB Split-Gal4 line, SS00117 | Janelia Research Campus | | *Wolff and Rubin, 2018* |
| Genetic reagent (*D. melanogaster*) | PB Split-Gal4 line, SS00090 | Janelia Research Campus | | *Wolff and Rubin, 2018* |
| Genetic reagent (*D. melanogaster*) | PB Split-Gal4 line, SS02198 | Janelia Research Campus | | *Wolff and Rubin, 2018* |
| Genetic reagent (*D. melanogaster*) | PB Split-Gal4 line, SS54549 | Janelia Research Campus | | *Wolff and Rubin, 2018* |
| Genetic reagent (*D. melanogaster*) | PB Split-Gal4 line, SS52577 | Janelia Research Campus | | *Wolff and Rubin, 2018* |
| Genetic reagent (*D. melanogaster*) | PB Split-Gal4 line, SS02239 | Janelia Research Campus | | *Wolff and Rubin, 2018* |
| Genetic reagent (*D. melanogaster*) | PB Split-Gal4 line, SS27583 | Janelia Research Campus | | *Wolff and Rubin, 2018* |
| Genetic reagent (*D. melanogaster*) | PB Split-Gal4 line, SS02303 | Janelia Research Campus | | *Wolff and Rubin, 2018* |
| Genetic reagent (*D. melanogaster*) | PB Split-Gal4 line, SS00078 | Janelia Research Campus | | *Wolff and Rubin, 2018* |
| Genetic reagent (*D. melanogaster*) | PB Split-Gal4 line, SS52257 | Janelia Research Campus | | *Wolff and Rubin, 2018* |

*Continued on next page*

*Continued*

| Reagent type (species) or resource | Designation | Source or reference | Identifiers | Additional information |
|---|---|---|---|---|
| Genetic reagent (*D. melanogaster*) | PB Split-Gal4 line, SS02195 | Janelia Research Campus | | ***Wolff and Rubin, 2018*** |
| Genetic reagent (*D. melanogaster*) | PB Split-Gal4 line, SS52244 | Janelia Research Campus | | ***Wolff and Rubin, 2018*** |
| Genetic reagent (*D. melanogaster*) | PB Split-Gal4 line, SS04778 | Janelia Research Campus | | ***Wolff and Rubin, 2018*** |
| Antibody | Anti-GFP (rabbit polyclonal) | Molecular Probes | Cat# A-21312, RRID:AB_221478 | (1:500) |
| Antibody | Anti-GFP (mouse monoclonal) | Thermo Fisher Scientific | Cat# A-11120, RRID:AB_221568 | (1:500) |
| Antibody | Anti-RFP (rabbit polyclonal) | Rockland | Cat # 600-401-379, RRID:AB_2209751 | (1:500) |
| Antibody | anti-BRP (mouse monoclonal) | DSHB | Cat# nc82, RRID:AB_2314866 | (1:150) |
| Antibody | Alexa Fluor 488 anti-rabbit (goat polyclonal) | Thermo Fisher Scientific | Cat# A11008, RRID:AB143165 | (1:1000) |
| Antibody | Alexa Fluor 568 anti-rabbit (goat polyclonal) | Thermo Fisher Scientific | Cat# A11011, RRID:AB_143157 | (1:1000) |
| Antibody | Cy5 anti-mouse (goat polyclonal) | Thermo Fisher Scientific | Cat# A10524, RRID:AB_2534033 | (1:1000) |
| Chemical compound, drug | Tryptone | VWR | 97063–386 | |
| Chemical compound, drug | 3-Iodo-L-tyrosine (3IY) | Sigma | I8250-5G | |
| Software, algorithm | SleepLab | William Joiner | | MATLAB-based software |
| Software, algorithm | FIJI | FIJI | | |
| Software, algorithm | Prism 8 | GraphPad | | |
| Other | USB webcam | LOGITECH | Logitech Webcam Pro 9000 | |
| Other | *Drosophila* Activity Monitoring (DAM) System | Trikinetics, Waltham, MA | | |

## Nutritional manipulations

Unless otherwise stated, flies were raised on standard food (described in detail below) in a 12 hr:12 hr light:dark (LD) cycle. Except where noted, 1- to 2-day-old flies in groups of 16 males and 16 females were transferred to normal food (standard food for *Drosophila* maintenance), 5% w/v sucrose-2% w/v agar food (yeast deprivation), or 5% w/v sucrose-2% w/v agar food supplemented with 2.75% w/v yeast extract (Fisher Scientific, Waltham, MA), 2.75% w/v tryptone (VWR, Radnor,

PA), or additional 2.75% sucrose. Flies were kept in these conditions for 6 days, with food renewed every 3 days. Standard food was composed of 6.56% w/v cornmeal, 2.75% w/v yeast, 0.81% w/v agar, 6.48% v/v molasses, 0.93% v/v propionic acid and 0.25% v/v tegosept (anti-fungal agent, Genesee Scientific, El Cajon, CA). In MF pairs, males of varying nutritional conditions were paired with 3- to 4-day-old females kept under the normal food condition, except where noted.

## Sleep analysis

Flies were raised and monitored at 25℃ except where noted. For sleep analysis, 4- to 8-day-old flies entrained to a 12 hr:12 hr LD cycle were placed in glass tubes containing 5% sucrose and 2% agar, with the following exceptions: in the Normal Food → Normal Food and Yeast Deprivation → Normal Food conditions in *Figure 2F–H*, and in Normal Food conditions in *Figures 4* and *6*, flies were placed in tubes containing normal food. For experiments involving TrpA1, flies were raised in LD at 22℃ and monitored for ~1.5 days at 22℃ to determine baseline levels, 1 day at 28-30℃ to activate the TrpA1 channel, and 1 day at 22℃ to examine recovery. Activity data were collected in 1 min bins using *Drosophila* Activity Monitoring (DAM) System (Trikinetics, Waltham, MA). Single-beam monitors were used except where noted. Beam breaks from single-beam monitors with infrared (IR) detectors at a single location or inter-beam movements from multi-beam monitors with IR detectors at 17 locations were used. Sleep was defined as a period of inactivity lasting at least 5 min. For video recording, flies were loaded into 9 mm x 19 mm x 4 mm recording arenas containing 5% sucrose and 2% agar. A USB webcam (Logitech Webcam Pro 9000) and infrared LEDs for nighttime recordings were used as previously described (*Machado et al., 2017*).

For DAM data, sleep parameters were analyzed using a MATLAB-based software, SleepLab (William Joiner). For video data, sleep and courtship (see below) of individual flies were manually scored for the first 5 min of each hour between ZT 18–24. We categorized behavior into three states: sleep, courting, wake but not courting (locomotion, eating, grooming, and brief inactivity). Sleep was defined as periods of no visible movement for at least 5 min. If a fly showed immobility for <5 min at the beginning or end of the 5 min analysis window, we examined the behavior before or after the analysis window. If the inactive period belonged to a sleep episode (≥5 min inactivity), we counted it toward sleep time. If not, we counted it toward wake but not courting time. Scoring was blinded to the experimental condition.

## Analysis of courtship and mating behavior

For courtship assay, virgin iso31 male flies were collected, housed in groups of ~10 in standard food for 1–2 days, and transferred to either 5% sucrose food or standard food. Flies were kept in these conditions for 7 days, with food renewed every 3 days. Virgin iso31 females were kept on standard food for 4–5 days in groups of ~10. Courtship assays were performed between ZT1 and ZT4. A male and female were gently aspirated into a plastic mating chamber (15 mm diameter and 3 mm depth) containing 5% sucrose and 2% agar and were kept separated until a divider was removed ~10 min later. Flies were recorded for 2 hr using a USB webcam (Logitech Webcam Pro 9000) and scored blind to experimental condition. Courtship index was determined as the fraction of total time a male was engaged in courtship activity during a 10 min period or until successful copulation after courtship initiation. Courtship activity included orienting, chasing, singing, and attempted copulation. For simultaneous analysis of courtship and sleep during the night, videos recorded under infrared light were manually scored for courtship and sleep during 5 min periods as described above.

## Immunohistochemistry

For whole mount immunohistochemistry, fly brains were fixed in 4% paraformaldehyde (PFA) for 30 min, dissected, and blocked in 5% normal goat serum for 1 hr at RT. Primary and secondary antibodies were incubated at 4℃ overnight. The following primary antibodies were used: rabbit anti-GFP (Molecular Probes, Eugene, OR, Cat# A-21312, RRID:AB_221478) at 1:500; mouse anti-GFP (Thermo Fisher Scientific, Waltham, MA, RRID:AB_221568) at 1:500; rabbit anti-RFP (Rockland Cat, Limerick, PA, # 600-401-379, RRID:AB_2209751) at 1:500; and mouse anti-BRP (DSHB, Iowa City, IA, Cat# nc82, RRID:AB_2314866) at 1:150. The secondary antibodies, Alexa Fluor 488 goat anti-rabbit (Thermo Fisher Scientific, Waltham, MA, Cat# A11008, RRID:AB143165), Alexa Fluor 568 goat anti-rabbit (Thermo Fisher Scientific, Waltham, MA, Cat# A11011, RRID:AB_143157), and Cy5 goat anti-

mouse (Thermo Fisher Scientific, Waltham, MA, Cat# A10524, RRID:AB_2534033) were used at 1:1000. Images were obtained on a Leica SP8 confocal microscope.

### Calcium imaging

Four to 7-day-old flies entrained to LD cycles were anesthetized on ice and dissected in adult hemolymph-like saline (AHL, 108 mM NaCl, 5 mM KCl, 2 mM CaCl2, 8.2 mM MgCl2, 4 mM NaHCO3, 1 mM NaH2PO4, 5 mM trehalose, 10 mM sucrose, 5 mM HEPES, pH 7.5, 265 mOsm; *Wang et al., 2003*). Dissected brains were mounted on a glass-bottom chamber containing AHL solution. A custom-built gravity-dependent perfusion system coupled to a flow valve (Warner Instruments, Hamden, CT) was used to control perfusion flow. Leica SP8 confocal microscope was used to acquire eight slices (~2.5 µm/slice) of the protocerebral bridge region every 5 s for 5 min. 2.5 mM ATP in AHL was delivered for 1 min after 1 min of baseline measurements. Average intensity projections were computed in FIJI, and the fluorescence intensity of the PB area was quantified. To correct for photobleaching of the fluorescence signal, frames acquired 30–60 s before the application of ATP and 150–180 s after the end of ATP perfusion were used to fit an exponential decay function. This fitted curve was then subtracted from the raw data, and the detrended data were used for subsequent analysis. The average intensity during the 30 s period prior to ATP perfusion was used as the baseline measurement, $F^0$. For each time point, normalized $\triangle F$, $(F-F^0)/F^0$, was computed.

### Statistical analysis

All analyses were performed using Prism 8 (GraphPad, San Diego, CA). To compare multiple groups, one-way ANOVAs were performed followed by Tukey or Dunnett T3 post-hoc tests depending on the type of pairwise comparisons. For experiments involving two factors, two-way ANOVAs were performed to test for the interaction, and a Sidak or Dunnett post-hoc test was employed to compare specific pairs of groups. Student's *t* test was used to compare pairs of groups. Brown-Forsythe and Welch's correction for unequal variances were employed when appropriate. Log-rank tests were used for cumulative courtship initiation and copulation success rates in mating assays. D'Agostino and Pearson tests were used to test for normality. Non-normally distributed data were analyzed by Mann-Whitney tests. All experiments were repeated on at least two separate occasions using flies from independent genetic crosses.

## Acknowledgements

We thank Drs. David Anderson, Liqun Luo, Amita Sehgal, Tanya Wolff and Gerald Rubin and the Bloomington Stock Center for fly stocks; Dr. William Joiner for the SleepLab software; Jennifer Wilson for suggestions for improving the manuscript; Kyle Kennedy, Joseph Buchler, Oghenerukevwe Akpoghiran, and Benjamin Peter Jenny for technical assistance; and members of the Koh lab for helpful discussions on the project. This work was supported by a Pew Latin American Fellowship (to JMD), a predoctoral fellowship from the Portuguese Foundation for Science and Technology (SFRH-BD-52321–2013 to DRM), and a grant from the National Institute of Neurological Disorders and Stroke (R01NS109151 to KK).

## Additional information

### Funding

| Funder | Grant reference number | Author |
| --- | --- | --- |
| Pew Charitable Trusts | Latin American Postdoctoral Fellowship | Jose M Duhart |
| National Institute of Neurological Disorders and Stroke | R01NS109151 | Kyunghee Koh |
| Portuguese Science and Technology Foundation | SFRH-BD-52321-2013 | Daniel R Machado |

The funders had no role in study design, data collection and interpretation, or the decision to submit the work for publication.

## Author contributions

José M Duhart, Conceptualization, Funding acquisition, Investigation, Methodology, Writing - original draft; Victoria Baccini, Yanan Zhang, Investigation, Writing - review and editing; Daniel R Machado, Funding acquisition, Investigation, Writing - review and editing; Kyunghee Koh, Conceptualization, Supervision, Funding acquisition, Investigation, Methodology, Writing - original draft, Writing - review and editing

## Author ORCIDs

José M Duhart (ID) https://orcid.org/0000-0002-7652-9707
Kyunghee Koh (ID) https://orcid.org/0000-0003-0847-8204

## Decision letter and Author response

Decision letter https://doi.org/10.7554/eLife.60853.sa1
Author response https://doi.org/10.7554/eLife.60853.sa2

# Additional files

## Supplementary files

• Transparent reporting form

## Data availability

All data generated during this study are included in the manuscript and supporting files.

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
