## [Decision Letter]

Thank you for submitting your article "Modulation of sleep-courtship balance by nutritional status in *Drosophila*" for consideration by *eLife*. Your article has been reviewed by three peer reviewers, and the evaluation has been overseen by Leslie Griffith as Reviewing Editor and Catherine Dulac as the Senior Editor. The following individual involved in review of your submission has agreed to reveal their identity: Ravi Allada (Reviewer #2).

The reviewers have discussed the reviews with one another and the Reviewing Editor has drafted this decision to help you prepare a revised submission.

We would like to draw your attention to changes in our revision policy that we have made in response to COVID-19 (https://elifesciences.org/articles/57162). Specifically, we are asking editors to accept without delay manuscripts, like yours, that they judge can stand as *eLife* papers without additional data, even if they feel that they would make the manuscript stronger. Thus the revisions requested below address clarity, data analysis and presentation.

Summary:

Animal have many needs (feeding, reproduction, sleep) and fulfilling these needs simultaneously is not always possible. How the brain makes choices between sleep/wake, feast/fast, and mate/abstinence is poorly understood. The authors tackle this problem using *Drosophila*. They and others previously found that reproductive drive can reduce sleep in males. They now look at the effect of nutrition on the interaction between reproductive behavior and sleep. They show that poor nutrition overwhelm the effects of females on male sleep behavior. They go on to map the neural circuitry mediating the nutritional effects on sleep/mating trade off. Combining anatomical with functional tools, they identify circuit elements downstream of courtship neurons.

Essential revisions:

The manuscript is well written and the figures clear and easy to interpret. Placing both main and supplementary figures at their relevant positions in the text, made reviewing a lot less of a hassle. There were two main areas of concern that the authors should deal with to improve the study, one that requires reanalysis of data and one that requires additional discussion.

1) There was concern about the analysis of data in Figure 4 and several of the supplementary figures. In Figure 4 nighttime sleep is quantified and compared at 30 C. This means that activation and genetic effects are confounded. It would be best to quantify the difference in nighttime sleep (deltaSleep) for each genotype between 22 and 30 C and then compare them. It is particularly crucial in this experiment where it appears there is some effect of the genetic background but Figure 3—figure supplements 1 and 2 should also be re-analyzed this way.

This reanalysis may lead to changes in conclusions. In the panel 4G, normal food condition, it is not clear what the asterisks on top of the DA-PB>TrpA1 condition represent. A difference relative to both parental controls, to one of them? In either case, the parental controls are very disparate which renders the result inconclusive rather than indicating that there is no sleep loss. Should, after the re-analysis mentioned suggested above the parental controls differ from the test condition in different directions, the text should be adjusted to reflect that.

2) The reviewers felt that there should be more discussion about the specifics of the food effects. The tryptone experiment leads the authors to conclude that "These data demonstrate that the balance between sleep and courtship in male flies is modulated by protein in dietary yeast". However, it is not clear they can exclude the possibility that the effect has to do with caloric density regardless of the source of calories. This should be mentioned and conclusions tempered. Reviewers were also interested in hearing more in the Discussion about the functional impact of yeast deprivation on reproduction and why the effects are modest, given the effects on sleep.

---

## [Author Response]

Essential revisions:The manuscript is well written and the figures clear and easy to interpret. Placing both main and supplementary figures at their relevant positions in the text, made reviewing a lot less of a hassle. There were two main areas of concern that the authors should deal with to improve the study, one that requires reanalysis of data and one that requires additional discussion.1) There was concern about the analysis of data in Figure 4 and several of the supplementary figures. In Figure 4 nighttime sleep is quantified and compared at 30 C. This means that activation and genetic effects are confounded. It would be best to quantify the difference in nighttime sleep (deltaSleep) for each genotype between 22 and 30 C and then compare them. It is particularly crucial in this experiment where it appears there is some effect of the genetic background but Figure 3—figure supplements 1 and 2 should also be re-analyzed this way.This reanalysis may lead to changes in conclusions. In the panel 4G, normal food condition, it is not clear what the asterisks on top of the DA-PB>TrpA1 condition represent. A difference relative to both parental controls, to one of them? In either case, the parental controls are very disparate which renders the result inconclusive rather than indicating that there is no sleep loss. Should, after the re-analysis mentioned suggested above the parental controls differ from the test condition in different directions, the text should be adjusted to reflect that.

We are thankful for the reviewers’ suggestions and have reanalyzed all experiments involving temperature change by quantifying delta-sleep. As the original DA-PB activation data were inconclusive when delta-sleep was analyzed, we performed further experiments. In the original data, the parental controls showed strong sleep suppression at 30ºC, potentially masking the arousing effects of DA-PB activation. Therefore, we used a lower temperature (29ºC) to activate TrpA1 and kept the flies at 29ºC for two days to compensate for the potentially lower TrpA1 activation levels. DA-PB resulted in significantly greater sleep loss on the 2^nd^ night at 29ºC compared with both parental controls in normally fed males but not in yeast-deprived males (new Figure 6).

Regarding Figure 4G (new Figure 6G), the asterisk was meant to indicate that DA-PB females were significantly different from both parental controls. We now explicitly state that sleep in the experimental females was different from both controls, but in opposite directions (subsection “DA-PB neurons regulate male sleep in a sex- and nutrition-dependent manner”).

2) The reviewers felt that there should be more discussion about the specifics of the food effects. The tryptone experiment leads the authors to conclude that "These data demonstrate that the balance between sleep and courtship in male flies is modulated by protein in dietary yeast". However, it is not clear they can exclude the possibility that the effect has to do with caloric density regardless of the source of calories. This should be mentioned and conclusions tempered. Reviewers were also interested in hearing more in the Discussion about the functional impact of yeast deprivation on reproduction and why the effects are modest, given the effects on sleep.

We appreciate the concern and performed additional experiments to address it. We show that 7.75% sucrose food, which is equivalent to 5% sucrose + 2.75% tryptone in caloric content, does not alter nighttime sleep in MF pairs compared to 5% sucrose (updated Figure 1K, L). This result further supports the idea that dietary protein modulates the balance between sleep and courtship.

Regarding the impact of yeast deprivation on reproduction, we are aware of two studies that examined the effects of dietary yeast on male reproductive fitness. Zajitschek et al., 2012, and Fricke et al., 2008, tested the effects of increasing dietary yeast content on male fitness, measured as the number of lifetime offspring (Zajitschek et al.) or as the paternity share of the males when females were allowed to copulate with other males (Fricke et al.). Fricke et al. found either no effect of yeast proportion or a bell-shaped dose response, depending on the experimental design. On the other hand, Zajitschek et al. found a negative correlation between dietary yeast content and the number of offspring. However, these studies focused on varying yeast content and did not assess the effects of a yeast-free diet. In our research, yeast deprivation substantially impacted nighttime sleep and courtship, but little influence on daytime courtship and mating. These findings suggest that yeast-deprived males are willing and able to perform reproductive behavior when awake (they are generally awake between ZT 1 and ZT 4). At night, flies are generally asleep, and yeast-deprived males may choose not to sacrifice their sleep to pursue female partners since the courtship is unlikely to result in viable progeny. We revised the Discussion section to incorporate these considerations (Discussion, second paragraph).